# Assessment of root phenotypes in mungbean mini-core collection (MMC) from the World Vegetable Center (AVRDC) Taiwan

**Muraleedhar S. Aski**[1]*, **Neha Rai**[1], **Venkata Ravi Prakash Reddy**[1], **Gayacharan**[2], **Harsh Kumar Dikshit**[1], **Gyan Prakash Mishra**[1], **Dharmendra Singh**[1], **Arun Kumar**[3], **Renu Pandey**[4], **Madan Pal Singh**[4], **Aditya Pratap**[5], **Ramakrishnan M. Nair**[6], **Roland Schafleitner**[7]

1 Division of Genetics, ICAR-Indian Agricultural Research Institute, New Delhi, India, 2 Germplasm Evaluation Division, ICAR-National Bureau of Plant Genetic Resources, New Delhi, India, 3 Division of Seed Science and Technology, ICAR-Indian Agricultural Research Institute, New Delhi, India, 4 Division of Plant Physiology, ICAR-Indian Agricultural Research Institute, New Delhi, India, 5 Crop Improvement Division, ICAR- Indian Institute of Pulses Research, Kanpur, India, 6 World Vegetable Center, South Asia/Central Asia, Patancheru, Hyderabad, India, 7 World Vegetable Center Taiwan, Molecular Genetics, Flagship Leader, Vegetable Diversity & Improvement Shanhua, Tainan, Taiwan

* murali2416@gmail.com

**Data Availability Statement:** All relevant data are within the paper and its Supporting Information files.

## Abstract

Mungbean (*Vigna radiata* L.) is an important food grain legume, but its production capacity is threatened by global warming, which can intensify plant stress and limit future production. Identifying new variation of key root traits in mungbean will provide the basis for breeding lines with effective root characteristics for improved water uptake to mitigate heat and drought stress. The AVRDC mungbean mini core collection consisting of 296 genotypes was screened under modified semi-hydroponic screening conditions to determine the variation for fourteen root-related traits. The AVRDC mungbean mini core collection displayed wide variations for the primary root length, total surface area, and total root length, and based on agglomerative hierarchical clustering eight homogeneous groups displaying different root traits could be identified. Germplasm with potentially favorable root traits has been identified for further studies to identify the donor genotypes for breeding cultivars with enhanced adaptation to water-deficit stress and other stress conditions.

## Introduction

Worldwide, mungbean or green gram (*Vigna radiata*) is being cultivated on nearly 7 million hectares area [1]. Among six Asiatic *Vigna* species, *Vigna radiata* is the most widely distributed species [2]. It is a major grain legume and cash crop which is widely cultivated in South, East, and South East Asia and is also increasingly grown in South America and sub-Saharan Africa. It fits in many intense cropping systems due to its photo-insensitivity and short duration nature. Mungbean is rich in easily digestible proteins, carbohydrates, fibers, minerals, vitamins, antioxidants, and other phytonutrients [3–5], thus can be used as a potential crop for the

**Funding:** The author(s) received no specific funding for this work

**Competing interests:** The authors have declared that no competing interests exist.

mitigation of malnutrition [6]. Mungbean is consumed as grain e.g. in *dhal*, as a sprout, or in various other countless preparations. The yield potential of mungbean is about 2 tonnes per hectare, while average productivity is nearly 0.5 tonnes per hectare. The large yield gap is primarily due to their cultivation on marginal land, suboptimal crop management, and abiotic and biotic stresses [7–9].

Mungbean, in comparison to other pulse crops, is relatively heat and drought tolerant, but their production is still affected by severe abiotic stresses, such as low or high temperatures [10], insufficient or excessive water [11, 12], high salinity [10], low soil fertility [13] and polluted heavy metal-containing soils [14, 15] and ultraviolet-B (UV-B) radiation [16].

Mungbean is mainly grown in three seasons in the Asian continent which is spring (February/March), summer (March/April) and *kharif* (June/July). Erratic water supply during these months exposes the seedlings to water stress when grown in rainfed conditions. Scarcity of water imposes stress at any plant stage [17]. Yields in tropical and subtropical countries such as India, Pakistan, and Ethiopia, will decline due to an expected higher incidence of water deficits [18]. The expansion of the global drought-restrained zone is threatening the overall mungbean crop production [19]. Insufficient availability of water on its own is more critical than any other environmental trigger for the growth of mungbean [20]. Water scarcity during the seedling stage hinders the establishment of healthy seedlings and limits overall yield. Drought produces many devastating effects on plants by disrupting various plant activities, such as carbon assimilation, reduced turgor, enhanced oxidative damage, and modifications in leaf gas exchange, resulting in a reduction in yield [21]. Better water supply for plants is critical for boosting crop production despite water shortages [22].

Root system architecture is seen as the main factor for efficient water absorption and therefore for maintaining productivity in conditions of drought [23]. Root features are commonly known as the root system architecture (RSA), which refers to the form of the roots and their physical space. With its ability to obtain more water from the soil, a deeper and more proliferative RSA is also capable of avoiding water deficit conditions.

Roots begin to spread into deeper soil layers as plants encounter water deficit stress [24, 25]. The root diameter and distribution of root conductivity-regulating metaxyl vessels are also documented to provide drought tolerance in food grain legumes [26]. Thicker roots prefer to penetrate deeper into soil layers [27].

The key determinants of proliferative rooting are the initiation and elongation of the lateral root, which usually refers to the sum of the lateral root, root volume, root surface, and root length density. Proliferative and deeper roots have increased capacity for water absorption in water-deficient soils [28, 29]. The ideal root phenotypes under water deficit situations, in food grain legumes, include root surface area [30], root length [31], deeper and proliferative roots [32, 33] in soybean (*Glycine max* L.), root diameter in cowpea (*Vigna unguiculata* L.) [34], root length in pea (*Pisum sativum* L.) [35], basal root angle in common bean (*Phaseolus vulgaris* L.) [36], and rooting depth, root surface area, root length density [37] and proliferative & deeper roots [38, 39] in chickpea (*Cicer arietinum* L.). Other crops that have benefited from ideal root phenotypes like proliferative and deep root traits under drought tolerance include rice (*Oryza sativa*) [40, 41], maize (*Zea mays*) [42, 43], barley (*Hordeum vulgare*) [44], and wheat (*Tritcum aestivum)* [45, 46].

Changing the root system architecture can improve desirable agronomic attributes such as yield, drought tolerance, and tolerance to nutrient deficiencies [47–49], and incorrect phenotyping and small mapping population sizes hinder the use of genomics to improve root characteristics in breeding programs [50]. To translate current physiological and genetic breakthroughs into improvements of yield and productivity especially in dry ecosystems, precise phenotyping and evaluation of root-related traits are vitally important. An effective

approach for increasing adaptation to edaphic stress is the selection and breeding of cultivars with root systems that more effectively use nutrients and water than current varieties [51].

To examine root features, several phenotyping techniques were documented, particularly hydroponic systems utilizing growth bags (or germination sheet) [52–54], agar-plate and aeroponic systems [55], soil rhizotrons [56–58], deep column methods [59], transparent containers [60], PVC pipes (columns) and glass-walled rhizoboxes filled with soil, but these methods are costly, require considerable labor and a large area for phenotyping larger genotype sets [61–63]. The unique semi-hydroponic phenotyping system [64] was established to assess the heterogeneity of the root trait in the narrow-leaf lupin (*Lupinus angustifolius* L.) core set [65, 66]. The same technique was used for root phenotyping in chickpea [67], maize, and barley [68]. This unique semi-hydroponic technique was modified to suit the purpose to screen the AVRDC mungbean mini-core collection for root traits.

The use of digital imaging and software tools for root image analysis is an innovative and efficient way to accurately assess root traits [69–71]. Several software programs are available to extract two-dimensional root morphology traits. This varies from the manual root label DART [70], commercially available software and semi-automated root analysis tool WinRhizo$^{TM}$ (Regent Instruments, Québec, Canada) (Pro, 2004) and EzRhizo [72], freely usable, fully integrated and automated SmartRoot [73] software for small root systems.

The World Vegetable Center (WorldVeg) has established a mungbean mini-core set, which represents a large proportion of the diversity available for this species in the WorldVeg gene bank [74] This resource comprises a major genetic resource for identifying new traits for future use in breeding programs. Characterizing the genotypic variability of the biodiverse accessions of the AVRDC mungbean mini-core collection for variation of root characteristics is the first step for identifying root traits for use in breeding more water and nutrient-use efficient varieties. The present study determined the genotypic variation of root characteristics in the AVRDC mungbean mini-core collection using a modified semi-hydroponic system and resulted in the grouping of the germplasm based on key root traits.

## Materials and methods

### Experimental materials and growth conditions

Plant material for this analysis included the WorldVeg mini-core collection of 296 genotypes [74] collected from 18 countries around the globe. The seeds were procured from the National Plant Genetic Resources Bureau (NBPGR), New Delhi (**S1 Table**). The obtained seed was multiplied in the field during the 2018 rainy season at the Indian Agricultural Research Institute (IARI) in New Delhi (Latitude 28° 38' 31.9236" N and Longitude 77° 9' 16.434"). The climate-controlled growth chamber [CONVIRON, Canada, PGW 36 with a growth area of 3.3 m$^2$ (36 ft$^2$)] at the National Phytotron Facility (NPF) in IARI, New Delhi, India was used for experiments. The studies were performed between September 2019 and December 2019. The day/night temperature of 30/18°C, 12 h photoperiod, and 90% relative humidity were maintained in the growth chamber. Seeds were surface-sterilized for 3 minutes in 0.1% HgCl$_2$ and then rinsed in double-distilled water before being kept for germination in a modified semi-hydroponics system.

### Modified semi-hydroponic system

To suit the experimental purpose, the semi-hydroponic technique [64] has been modified. Our fundamental goal was to screen larger germplasm sets for a shorter time. This was accomplished by changing the bin size with smaller plastic trays and germination stands, resulting in a 26 cm long germination stand with 12 cm width and 8 cm height (specifically created by Bio-Link Pvt.

Ltd. New Delhi). The size of the plastic tray was 51 cm in length, 43 cm in width, and 13 cm in height (Tarson products Pvt. Ltd.). Germination paper (SGPK-145; GSM with the creepy surface was obtained from Bio-Link Pvt. Ltd New Delhi) with a size of 14 x 8 cm was used. Two equally cut germination paper sizes (length 14 cm x height 10 cm) were positioned in each germination cell to accommodate seeds for germination. Eight liters of double distilled water were used to moisten the germination paper stand in the plastic tray. For each accession, 10 seeds were used for germination in each germination cell. In each germination stand, 12 germplasm accessions were housed (**Fig 1**). Three germination stands were placed in each plastic tray, to grow thirty-six mungbean accessions. A total of 9 plastic trays, each containing 36 accessions, covered 296 AVRDC mungbean mini-core collection entries. Due to hard seeds produced during seed multiplication in the field, we were at risk of poor germination. To prevent variable germination rate, uniform 10 seeds were used for germination [75]. Three uniform, healthy seedlings were selected from the emerging seedlings as three technical replicates after 20 days.

Each tray was filled with 8 liters of modified Hoagland solution when cotyledon leaves were developed. The basic nutrient solution consisted of 0.92 mM $K_2SO_4$, 1 mM $MgSO_4$, 5 mM urea, 0.75 mM $CaCl_2.2H_2O$, 0.04 mM Fe-EDTA) and micronutrients (0.62 µM $CuSO_4$), 0.6 µM $ZnSO_4$, 2.4 µM $H_3BO_3$, 0.6 µM $Na_2MoO_4$ and 0.9 µM $MnSO_4$ [76]. The pH of the nutrient solution was maintained at 6.0 with 1 M HCl or 1 M KOH for adjustment. The solution in the trays was replaced on alternating days and the entire system was periodically aerated by small aquarium air pumps (SOBOTM TM, 5W, 2-way air pump with 2 nozzles, 4.2 W, and 2 x 5.5 L, output power).

## Root scanning for image capture

To capture root images, twenty days old seedlings were used. The intact root system was harvested from each plant and carefully spread, without overlapping roots, over a scanning tray of

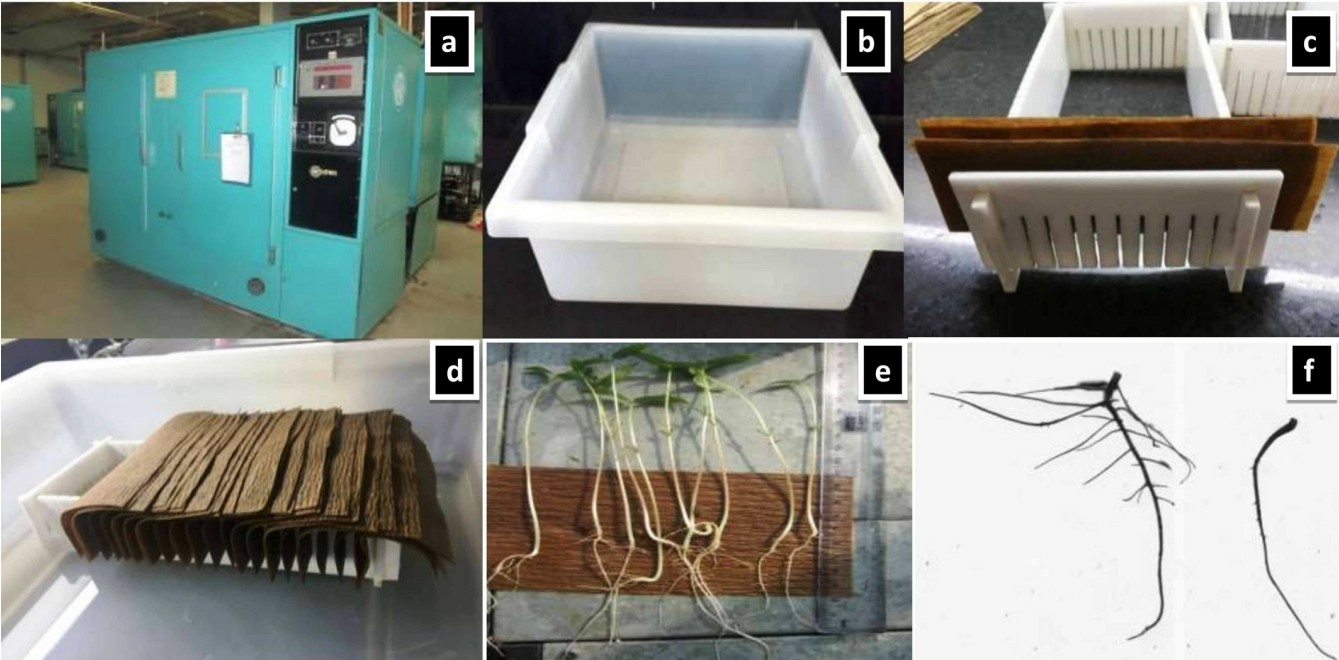

**Fig 1.** Semi-hydroponic phenotyping platform (a) growth chamber (b) plastic tray (c) germination stand and paper setup (d) germinating mungbean seeds in semi-hydroponic phenotyping platform (e) seedling grown for 18 days (f) root system of two contrasting genotypes grown for 18 days.

a root scanner (EPSON$^{TM}$ V700). TIFF-format grayscale quality images were analyzed by WinRhizo$^{TM}$ (Pro version 2016; Regent Instrument Inc., Quebec, Canada). Setup parameters: image resolution of 400 dpi, manual dark root on white background, scanner calibration, 8-bit depth, image resolution of 4395 x 6125 pixels, 0 mm focal length. Roots are distributed in a 30 x 40 x 2 cm acrylic tray with a volume of 700 ml water. Manually, the debris was separated from the sample roots by suspension in a beaker containing water. The trash-free clean roots were used for scanning.

## Root image analysis

Total root length (TRL), total surface area (TSA), primary root length (PRL), total root volume (TRV), root average diameter (RAD), total root tips (TRT), total root forks (TRF), and total root crossings (TRC)were the major root traits analyzed by the WinRhizo$^{TM}$ software. PRL was measured manually through a steel measurement scale (EISCO-GROZTM).

The WinRhizo$^{TM}$ provided main and additional data to classify total root length (TRL), total root volume (TRV), total root surface area (TSA), and total root tips (TRT) into five classes of root diameter intervals: class1: (0–0.5 mm), class 2: (0.5–1.0 mm), class 3: (1.0–1.5 mm), class 4: (1.5–2.0 mm) and class 5: (>2.0 mm) [77–79]. In every class, the root traits were calculated as a proportion of the total trait [68]. The details of each tested trait are given in **Table 1**. Biomass related traits like root dry weight (RDW) (mg), shoot dry weight (SDW) (mg), and total dry weight (TDW) (mg) were determined using a digital weighing balance (Citizen$^{TM}$, CX 265) on three biological replicates after air-forced drying in an oven at 70°C for 72 h.

## Statistical analysis

The data were subjected to descriptive and summary statistics like mean, standard deviation, skewness, kurtosis, coefficient of variation, and Pearson's correlation by STAR (Statistical Tool for Agricultural Research) 2.1.0 software [80]. Principal component analysis (PCA), frequency distribution, and normal curve fitting were performed using PAST 4.03 software [81].

Three distinct classes of the AVRDC mungbean mini-core accessions were categorised based on standard deviation (SD) and mean ($\overline{x}$): (i) $\leq \overline{x}$ –SD, (small trait value) (ii) ($\geq \overline{x}$– SD) to ($\leq \overline{x}$ + SD), (average trait value) and (iii) $\geq \overline{x}$ + SD (high trait value) [82, 83]. For every root

**Table 1. Description of the root traits assessed in the study.**

| Abbreviated name | Full trait name | Description | Measurement description |
|---|---|---|---|
| PRL | Total Root Length (cm) | Average of primary root length in three plants | Measured using a scale |
| TPA | Total Project Area(cm$^2$) | Average of total project area in three plants | Measured electronically by WinRhizo Software |
| TSA | Total Root Surface Area (cm$^2$) | Average root surface area inthree plants | Measured electronically by WinRhizo Software |
| TRL | Total Root Length(cm) | Average root length of three plants | Measured electronically by WinRhizo Software |
| ARD | Average Root diameter (cm) | Average root diameter of three plants | Measured electronically by WinRhizo Software |
| LPV | Length Per Volume (cm/mm) | Average of length per volume in three plants | Measured electronically by WinRhizo Software |
| TRV | Total Root Volume (cm3) | Average root volume of three plants | Measured electronically by WinRhizo Software |
| TRT | Total Root Tips(Number) | Average of root tips inthree plants | Measured electronically by WinRhizo Software |
| TRF | Total Root Forks (Number) | Average of Total root forks in three plants | Measured electronically by WinRhizo Software |
| TRC | Total Root Crossings (Number) | Average of Total root crossings in three plants | Measured electronically by WinRhizo Software |
| RDW | Root dry weight in (mg) | Average Root dry weight in three plants | Measured manually in digital scale |
| SDW | Shoot dry weight (mg) | Average Shoot dry weight in three plants | Measured manually in digital scale |
| TDW | Total dry weight (mg) | Average Total dry weight in three plants | Measured manually in digital scale |
| RSR | Root to Shoot Ratio (mg/mg) | Average of Root to Shoot Ratio in three plants | Measured manually |

character the H', Shannon-Weaver diversity index [84–86] was determined using the formula:

$$H' = -\sum_{i=1}^{R} pi\,(\ln pi)$$

In which,

1. pi is the proportion of individuals belonging to the i[th] class

2. s is the total number of accessions.

## Results

### Phenotypic variation

The AVRDC mungbean mini-core collection displayed distinct phenotypic differences of the traits under investigation when grown in the modified semi-hydroponic system. The coefficient of variation of the traits observed was greater than 30% (**Table 2**). The root characteristics exhibiting high variation were PRL, TSA, TRL, TRF, and LPV. Trait PRL ranged from 133.38 cm (EC 862594) to 1.96 cm (IC 616154) and TRL ranged from 60.35 cm (EC 862670) to 0.79 cm (EC 862662). Only minor differences were found for TSA (16.26 cm² for IC 616276 to 1.04 cm² for EC 862662), ARD (1.74 cm for EC 862653 and 0.39 cm for IC616115) and TRV (0.19 cm³ for EC 862645 to 0.01 cm³ for IC 616114) (**Tables 2 and S2**).

The frequency distribution of most root and biomass traits was skewed towards a smaller trait value, except for PRL and TPA, which showed a near-normal distribution (**Fig 2**). Fine roots (less than 1 mm) made up the bulk of the root system in all genotypes, while accessions originating in Australia had the highest proportion of roots with a diameter of 1.00–1.5 mm (**Fig 3**).

### Correlation among root traits

All root traits were positively correlated with each other, except for ARD, which was negatively correlated with the other traits. The biomass traits RDW, SDW, TDW, and RSR did not show

**Table 2. Candidate trait variation in the AVRDC mungbean mini-core collection.**

| S.No | Traits | Max | Mini | Mean±SD | CV (%) | Skewness | Kurtosis |
|------|--------|------|------|----------|--------|----------|----------|
| 1 | PRL | 133.38 | 1.96 | 39.64±29.96 | 75.577 | 0.81 | -0.23 |
| 2 | TPA | 8.79 | 1.14 | 4.51±1.76 | 39.050 | 0.40 | -0.50 |
| 3 | TSA | 16.26 | 1.04 | 7.48±3.66 | 49.003 | 0.32 | -0.84 |
| 4 | TRL | 60.35 | 0.79 | 16.62±11.82 | 71.150 | 0.78 | 0.17 |
| 5 | ARD | 1.74 | 0.39 | 0.71±0.23 | 32.594 | *1.35* | *1.86* |
| 6 | LPV | 60.23 | 0.12 | 16.35±11.82 | 72.333 | 0.78 | 0.17 |
| 7 | TRV | 0.19 | 0.01 | 0.05±0.02 | 51.396 | *1.16* | *2.32* |
| 8 | TRT | 181.00 | 2.00 | 25.02±24.63 | 98.412 | *2.65* | *10.98* |
| 9 | TRF | 90.00 | 0.00 | 18.79±17.30 | 92.061 | 0.87 | 0.68 |
| 10 | TRC | 12.00 | 0.00 | 1.18±1.89 | 59.763 | *2.18* | *5.69* |
| 11 | RDW | 99.97 | 2.60 | 25.57±18.78 | 73.463 | *1.26* | *1.64* |
| 12 | SDW | 177.11 | 2.10 | 44.96±32.14 | 71.481 | *1.11* | *1.40* |
| 13 | TDW | 277.00 | 10.00 | 70.54±49.89 | 70.737 | *1.16* | *1.56* |
| 14 | RSR | 3.81 | 0.05 | 0.61±0.28 | 46.746 | *6.83* | *67.15* |

*Skewness and kurtosis were analyzed for seedling root traits, and values bigger than 1 are highlighted in italics

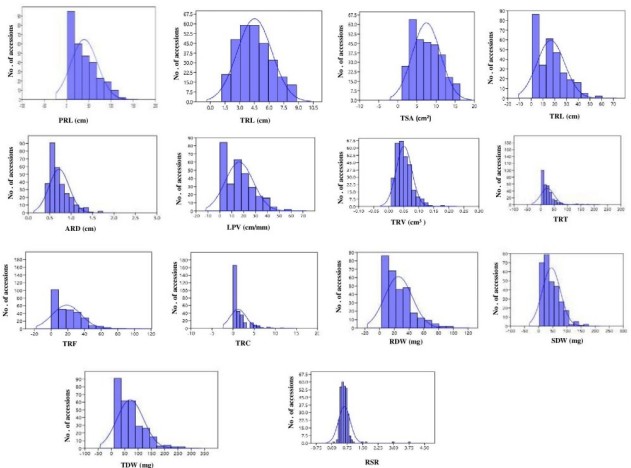

**Fig 2. Frequency distribution of root traits in the AVRDC mungbean mini core collection with the trait value on the x axis and the number of accessions on the y-axis.**

any association with root traits. While RDW showed positive association with other biomass traits like SDW and TDW (**Fig 4**).

## Principal components analysis for root and shoot trait variability

Fourteen characteristics were used in the PCA and 95.61 percent of the total variation was captured by three principal components (PCs) with eigenvalues > 1. For TRL, SDW, RDW, and

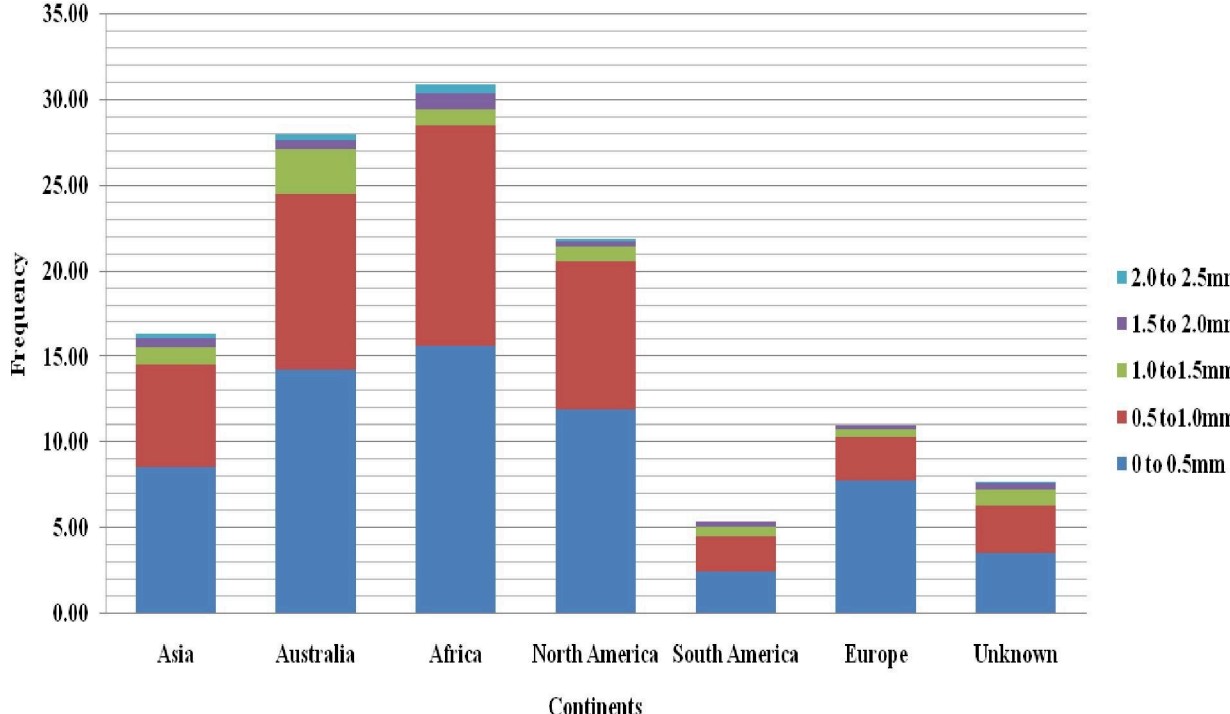

**Fig 3. Distribution of total root length based on root diameter classes (0 to 0.5mm, 0.5 to1.0mm, 1.0 to1.5mm, 1.5 to 2.0mm and 2.0 to 2.5mm).**

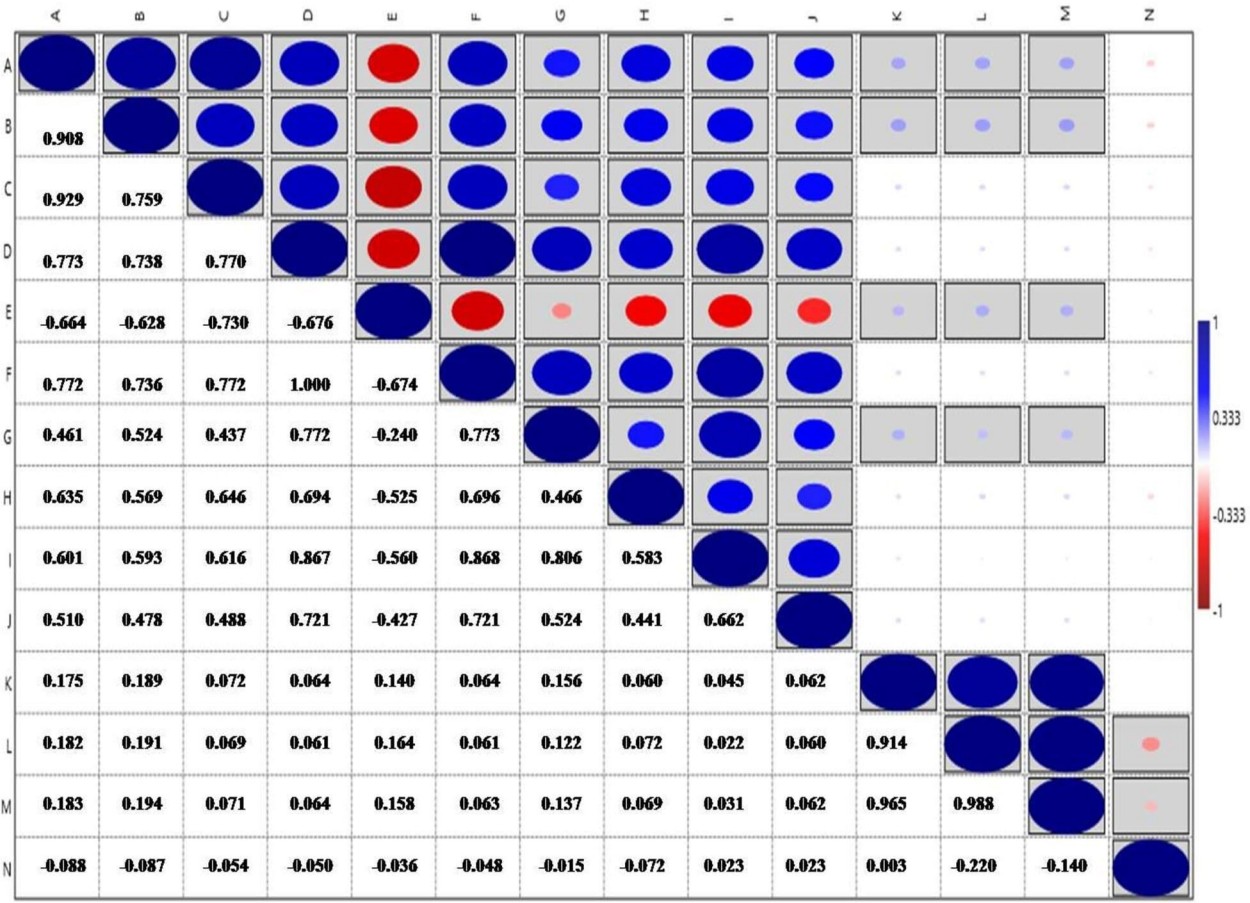

**Fig 4. Correlations coefficients among all the root and biomass traits, boxed blocks indicate significant correlations at (p<0.05).** (A: PRL, B: TPA, C: TSA, D: TRL, E: ARD, F: LPV, G: TRV, H: TRT, I: TRF, J: TRC, K: RDW, L: SDW, M: TDW, N: RSR).

TDW, PC1 accounted for 64.93 percent of the variability. PC2 accounted for 26.34% of the variability contributed by PRL, LPV, TRF, and TRT (**Table 3**). Biplots (**Fig 5A and 5B**) display the distribution of genotypes based on PCA regression scores, indicating the relative distance between the AVRDC mungbean mini-core accessions based on the combination of root trait values. 95.61 percent of the variability was expressed by loading plots. The PC1 vs PC2 biplot showed 14 genotypes as outliers (**Fig 5A**) and 21 genotypes were classified as outliers in the PC1 vs PC 3 biplot (**Fig 5B**). PCA loading scores showed that the characteristics TDW>SDW>RDW>PRL are major contributors with the magnitude of their contribution to PC1 in decreasing order, while PRL>TRT>TRF>TRL>LPV are contributing characteristics in PC 2 (**Table 4**).

## Diversity pattern and grouping by trait performance

The mungbean genotypes were classified into 3 groups, namely low, medium, and high trait diversity (**Table 4**). Most genotypes belonged to the medium group for all traits. The traits PRL, ARD, TRT, TRC, RDW, SDW, and RSR had a relatively large proportion of genotypes in the high trait value ($\geq \bar{x}$+SD) category, while for TPA, TRL, LPV, TRF, and TDW a greater number of genotypes were in the low trait value ($\leq \bar{x}$ –SD) category.

**Table 3. Different principal components and their loading scores trait wise.**

| | PC 1 | PC 2 | PC 3 | PC 4 | PC 5 | PC 6 | PC 7 | PC 8 | PC 9 | PC 10 | PC 11 | PC 12 | PC 13 | PC 14 |
|---|---|---|---|---|---|---|---|---|---|---|---|---|---|---|
| **PRL** | **0.130** | **0.671** | -0.632 | -0.317 | 0.026 | -0.138 | 0.073 | -0.024 | -0.085 | 0.003 | -0.009 | -0.010 | 0.001 | 0.000 |
| TPA | 0.008 | 0.036 | -0.034 | -0.007 | 0.002 | -0.003 | 0.311 | -0.261 | 0.906 | -0.011 | 0.034 | 0.098 | -0.010 | 0.000 |
| TSA | 0.009 | 0.082 | -0.059 | -0.027 | 0.010 | -0.004 | -0.820 | 0.403 | 0.381 | -0.023 | 0.050 | 0.080 | -0.004 | 0.000 |
| TRL | 0.027 | **0.266** | 0.012 | **0.295** | -0.084 | **0.575** | 0.004 | -0.071 | -0.073 | -0.180 | 0.660 | 0.157 | -0.007 | 0.000 |
| ARD | 0.000 | -0.004 | 0.002 | -0.001 | -0.001 | -0.004 | 0.027 | -0.005 | -0.107 | -0.003 | -0.238 | 0.963 | -0.068 | 0.000 |
| **LPV** | 0.027 | **0.266** | 0.014 | **0.295** | -0.082 | 0.574 | -0.050 | -0.043 | 0.043 | 0.186 | -0.664 | -0.154 | 0.005 | 0.000 |
| TRV | 0.000 | 0.000 | 0.000 | 0.001 | 0.000 | 0.000 | 0.001 | -0.003 | 0.002 | -0.003 | -0.008 | 0.069 | 0.998 | 0.000 |
| **TRT** | 0.056 | **0.514** | **0.764** | -0.381 | 0.033 | -0.057 | 0.012 | 0.003 | 0.001 | 0.000 | 0.001 | 0.000 | 0.000 | 0.000 |
| **TRF** | 0.027 | **0.338** | **0.106** | **0.757** | 0.025 | -0.547 | -0.004 | -0.010 | -0.007 | -0.002 | 0.000 | -0.001 | -0.001 | 0.000 |
| TRC | 0.003 | 0.029 | 0.001 | 0.050 | -0.012 | 0.067 | 0.472 | 0.873 | 0.089 | -0.019 | 0.002 | 0.002 | 0.002 | 0.000 |
| **RDW** | **0.288** | -0.041 | 0.004 | 0.048 | 0.756 | 0.090 | 0.004 | -0.001 | -0.003 | -0.016 | -0.004 | -0.001 | 0.000 | 0.577 |
| **SDW** | **0.509** | -0.073 | 0.024 | -0.020 | -0.629 | -0.071 | -0.008 | 0.002 | 0.004 | 0.014 | 0.003 | 0.000 | 0.000 | 0.577 |
| **TDW** | **0.797** | -0.115 | 0.028 | 0.028 | 0.126 | 0.019 | -0.003 | 0.001 | 0.000 | -0.002 | 0.000 | -0.001 | 0.000 | -0.577 |
| RSR | -0.001 | 0.000 | 0.000 | 0.002 | 0.022 | 0.000 | 0.004 | 0.018 | 0.000 | 0.965 | 0.251 | 0.065 | 0.000 | 0.000 |

## H'- Shannon-Weaver diversity index

Using the Shannon-Weaver diversity index (H'), the phenotypic diversity among the characters was compared. A high H' defined balanced frequency groups for a character and high diversity, while a low H' suggested an unbalanced frequency for a trait and low diversity. H' values for the traits were distinct and ranged from 0.37 to 0.96 between the genotypes (Table 4). Root traits including TSA, TRL, LPV, and TRF were more diverse than TRT, ARD, and TRC. For most of the characteristics, the diversity indices were above 0.5, suggesting the existence of sufficient heterogeneity. However, total root tips (0.36) and total root crossings (0.37) showed unbalanced frequency and lacked diversity.

## Diversity of the AVRDC mungbean mini-core collection based on root traits

Among the mini-core accessions, an agglomerative hierarchical clustering (AHC) dendrogram showed significant trait diversity (Fig 6). Entries were divided into eight clusters by cluster analysis. With 135 entries, cluster VI was the largest, and with just one entry from Iran, cluster VIII was the smallest (IC 862636). Clusters VII (78 entries) and V (55 entries) were the second and third-largest clusters. The second and third smallest clusters are Clusters IV (2 entries of Indian origin) and III (4 entries, two of Indian origin and one each from the Philippines and Australia).

## Discussion

Across both breeding programs and scientific research, the quest for root characteristics has intensified, offering improvement in the acquisition of resources and tolerance to abiotic stress including heat and drought stress, especially in resource-poor environments. Advancement was delayed due to problems in effective and accurate root trait phenotyping in large germplasm panels [49, 50, 87]. In the AVRDC mungbean mini-core collection of 296 genotypes, the use of a modified semi-hydroponic phenotyping system saved time and space for phenotyping and provided access to a substantial variation of root traits.

The novel method of semi-hydroponic phenotyping [64] has been modified to screen and explore the genetic variation of different root characters at the early vegetative stage of the

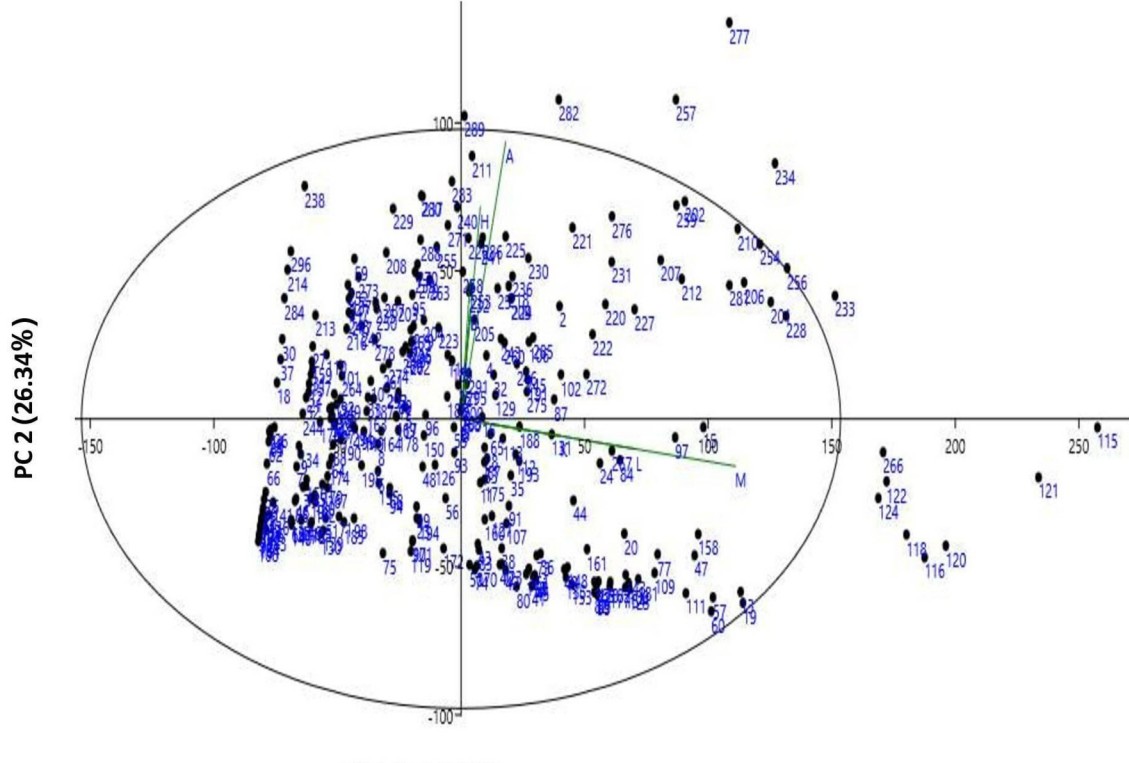

**PC 2 (26.34%)**

**PC 1 (64.94 %)**

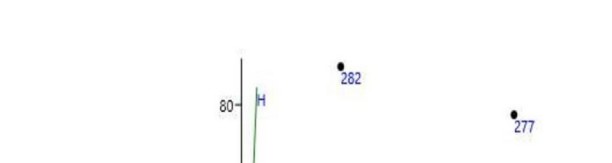

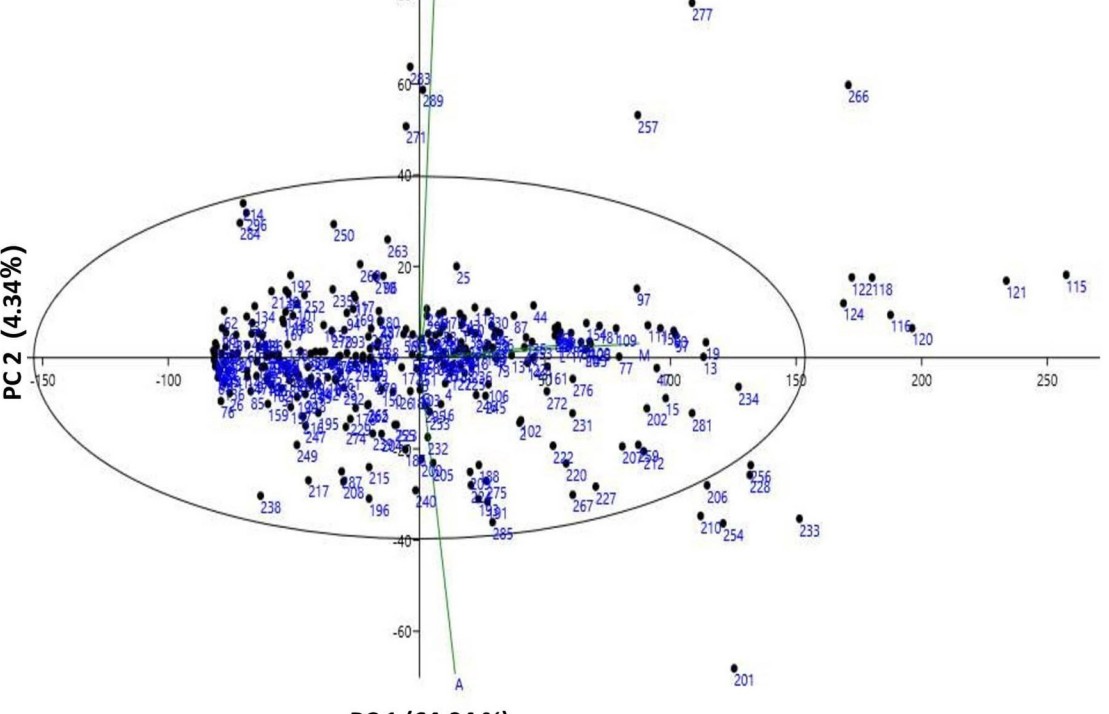

**PC 2 (4.34%)**

**PC 1 (64.94 %)**

**Fig 5.** Biplots and outliers in comparison between a) Principle Component 1 vs. Principle Component 2. Biplots and outliers in comparison between b) Principle Component 1 vs. Principle Component 3. (Where A: PRL, B: TPA, C: TSA, D: TRL, E: ARD, F: LPV, G: TRV, H: TRT, I: TRF, J: TRC, K: RDW, L: SDW, M: TDW, N: RSR).

AVRDC mungbean mini-core collection (**Tables 2 and S1**). Previously, the semi-hydroponic phenotyping platform provided quality data for the wild narrow-leaf lupin [65], core collections of lupin [66], and chickpea [67] to assess the genetic basis of variation in root characteristics.

The mungbean root system is similar to those of other dicotyledonous species including Arabidopsis, Medicago, and other legumes like chickpea and develops through consecutive branch/lateral root orders from a primary root that emerges from the embryo [88]. The volume and size of lateral branches/roots is an important contributor to the growth and development of food grain legumes [89].

Depth of rooting and density of root branches are essential architectural features that directly affect water and nutrients acquisition in the soil strata [90]. The PRL of the deep-rooting genotype EC862594 was more than twice as high as the average value of the whole germplasm set and genotype EC862670 had a more than 3-fold TRL than the average, corroborating the trait diversity present in the AVRDC mungbean mini-core collection (**S2 Table**). Another aspect that affects deep rooting is root penetrability and root thickness or root diameter [91]. The thicker roots prefer to go deeper into the soil to obtain water from deeper soil layers [27]. Research shows that in chickpea there is a strong association between the prolific root system and the development of grain in terminal drought situations [38]. In the case of total surface area, the best genotype IC616276 only had about 20% more TSA than the average. This suggests that the TSA exhibited lower variations in the AVRDC mungbean mini-core collection (**Tables 5 and S2**).

Proliferative rooting is primarily characterized by the initiation and elongation of lateral roots, which usually refers to the number of the lateral root, the root length density (RLD), and the root volume and the root surface area. Proliferative roots have a considerably large water

**Table 4. Candidate traits and their Shannon-Weaver diversity indices (H') in the AVRDC mungbean mini-core collection.**

| S.No | Trait | 296 mungbean lines | | | Shanon weaver index |
|------|-------|-----|--------|------|---------------------|
|      |       | Low | Medium | High |                     |
| 1    | PRL   | 48  | 195    | 53   | 0.88 |
| 2    | TPA   | 51  | 198    | 47   | 0.86 |
| 3    | **TSA** | 58 | 180  | 58   | **0.94** |
| 4    | **TRL** | 65 | 181  | 50   | **0.93** |
| 5    | ARD   | 23  | 227    | 46   | 0.69 |
| 6    | **LPV** | 65 | 181  | 50   | **0.93** |
| 7    | TRV   | 41  | 210    | 45   | 0.80 |
| 8    | TRT   | 0   | 262    | 34   | 0.36 |
| 9    | **TRF** | 79 | 170  | 47   | **0.96** |
| 10   | TRC   | 0   | 260    | 36   | 0.37 |
| 11   | RDW   | 30  | 223    | 43   | 0.73 |
| 12   | SDW   | 46  | 199    | 51   | 0.86 |
| 13   | TDW   | 45  | 209    | 42   | 0.81 |
| 14   | RSR   | 127 | 135    | 34   | 0.56 |

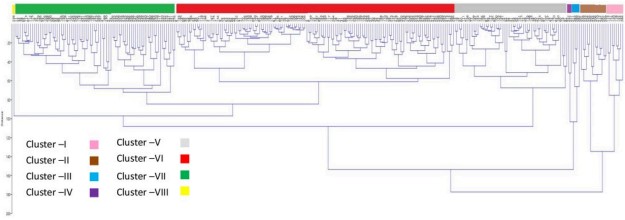

**Fig 6. Dendrogram depicting the diversity of the AVRDC mungbean mini-core accessions.** Clusters C-I(pink), C-II (brown), C-III(blue), C-IV(violet), C-V(grey), C-VI(red), C-VII(green) and C-VIII(yellow). (A separate picture file was uploaded for dendrogram).

absorption potential in water deficit soils. In environments with water scarcity, lines with higher RLD showed improved yield and drought-tolerance-related performance [92].

Mungbean is predominantly cultivated on soils with residual moisture from previous rainy seasons. Terminal drought stress, especially at the end of the growing season, is a major constraint restricting the yield of mungbean [8, 10, 93]. The mungbean genotypes with longer primary roots and larger total surface area could perform better in terms of water and nutrient acquisition, especially when water and nutrients are heterogeneously distributed across different soil levels. This research identified deeper rooting genotypes, like EC862594, IC616203, IC616109, IC616184, and EC862589 (**S2 Table**), that could access water from deeper soil layers whenever the topsoil dries up later during the season.

The root morphology traits vary from species to species and also between different genotypes of a species [65, 93, 94]. Root architectural features such as TSA, TRL, ARD, PRL, and TRV were responsible for most of the observed root trait variability at the seedling level. Correlation studies showed a positive correlation among TRL, PRL, TPA, TRV, TSA, TRT, and TRF. The stated traits also exhibited a negative correlation with the average root diameter (ARD). The selection of higher ARD values would negatively affect the root traits mentioned above. Plants with smaller root diameter and a specific root length of fine roots are found to be better suited to dry conditions [95]. The root surface area and root length are mainly influenced by the root diameter [96].

**Table 5. Variability of root architectural details of ten contrasting genotypes of the AVRDC mungbean mini-core collection grown under semi-hydroponic conditions.**

| | Top five genotypes having ideal phenotypes | | | | | | | | |
|---|---|---|---|---|---|---|---|---|---|
| No. | PRL | TSA | TRL | ARD | TRV | RDW | SDW | TDW | RSR |
| 1 | EC862594 | IC616276 | EC862670 | EC862653 | EC862645 | IC862615 | EC862617 | EC862617 | EC862602 |
| 2 | IC616203 | IC61625 | IC616247 | IC616222 | IC61625 | EC862617 | IC862615 | IC862615 | EC862588 |
| 3 | IC616109 | IC616175 | IC61625 | EC862661 | IC616247 | IC616200 | EC862654 | EC862654 | IC616169 |
| 4 | IC616184 | IC616107 | IC616166 | EC862651 | IC616208 | IC616150 | EC15198 | IC616200 | IC616148 |
| 5 | EC862589 | EC15046 | IC616101 | IC616191 | IC616118 | EC862654 | IC616150 | IC616150 | IC616106 |
| | Bottom five genotypes with undesirable phenotypes | | | | | | | | |
| No | PRL | TSA | TRL | ARD | TRV | RDW | SDW | TDW | RSR |
| 1 | IC616154 | EC862662 | EC862662 | IC616115 | IC616114 | IC862636 | EC862602 | EC862659 | EC862662 |
| 2 | EC862634 | EC862622 | EC862634 | EC862585 | IC616197 | IC616239 | EC862588 | IC616258 | IC616263 |
| 3 | EC862662 | EC862634 | IC616250 | IC616271 | EC15184 | EC862622 | EC862659 | EC15184 | IC616099 |
| 4 | EC862622 | EC862651 | IC616195 | EC862646 | IC616154 | EC862605 | EC862611 | IC616154 | IC616269 |
| 5 | EC862651 | IC616194 | IC616154 | IC616120 | EC862634 | IC616220 | IC616258 | EC15216 | IC616226 |

Water and nutrient uptake capability are determined by root architecture. In the early stages, genotypes with vigorous root growth (TRL and TSA) take up water and minerals more effectively and have better seedling establishment [97], resulting in increased photosynthetic ability, a higher output of biomass, and a higher survival rate under stressful conditions. The root number and TRL [98, 99] are positively correlated with yield and biomass.

Our analysis showed that among the diameter classes large parts of the mungbean root system consist of a variety of very fine and fine roots in diameters between 0.5 and 2.0 mm. The absorption of water and nutrients, the involvement of very fine and fine roots are well established [64, 100, 101]. In the root system, a high percentage of fine roots contributes to an improvement in TSA for the acquisition of more water and nutrients and helps plants to cope with stress [102].

The root to shoot ratio (RSR) is also used to predict the distribution of biomass among roots and shoots [103]. In rice seedlings, the water deficit situation raises the root-to-shoot ratio by altering enzymatic activity and carbohydrate balancing [104]. EC862602, EC862588, and IC616169 are genotypes identified with higher RSR values **(Tables 5 and S2).** In the phosphorus efficiency studies, higher ratios of root to shoots are often labeled as index traits because of the improvement in root biomass and the large deep root system required to extract more nutrients [105, 106].

The Cluster II, VI, and VIII genotypes are candidates for crossing programs to produce successful root trait recombinants such as PRL, TSA, TRL, TPA, and TRT. The cross-combinations (EC 862594 (C-II) x IC 616154(C-VI), TSA (IC 616154(C-II) x EC 862622(C-VI), TRL (EC 862670(C-VII) x EC 862662(C-VI), TPA(IC 15252(C-VII) x IC 616154(C-VI)) will help increase PRL. For TSA and TRV, genotypes in clusters V and III are diverse **(Fig 3).** The rate of absorption of nutrients is dependent on the TSA and TRV [107, 108]. The root traits TSA, TRL, and TRV were the target traits for mungbean to increase the efficiency of nutrient use (especially phosphorus) at the seedling stage [109]. Increasing nitrogen efficiency in maize, the RDW and TRL played an important role [110]. IC 862615, EC 862617, IC616200, and IC616150 were identified with having higher RDW **(Tables 5 and S2).** In the case of finger millet, the starvation reaction to phosphorus is mitigated by increased TRL and root hair count and length [111].

A thorough understanding of the multiple associations between root traits [112] is needed for the proper use of root traits in crop breeding. To demonstrate the relationship and diversity of the characteristics and relative homogeneous grouping of genotypes based on root traits, our research examined principal component analysis, hierarchical clustering, Pearson's correlation, and Shanon-Weaver diversity indices. The variations contained in the AVRDC mungbean mini-core collection for PRL, TSA, TPA, TRT, and TRF have the potential for mungbean improvement programs and genotype categorization in high trait value, medium trait value, and low trait value classes based on mean and SD facilitates the selection of breeding materials [113, 114].

The Shannon-Weaver diversity index (H') showed phenotypic diversity among the root characteristics, where low H' values stand for unbalanced frequency distribution and lack of trait diversity [115], while high H' means high genetic diversity in traits [116]. In the AVRDC mungbean mini-core collection, root characteristics i.e., TRL, TSA, LPV, and TRF had a relatively high level of H' (>0.9) showing high diversity for these characteristics. The high H' value and the positive correlation of these characteristics with PRL and TPA showed that these characteristics are appropriate at the seedling stage to improve water and nutrient uptake efficiency in mungbean. Besides, variations in ARD, TRV, and TRT would also be useful in stressful environments to increase nutrient productivity and crop yields [117].

The genetic and molecular basis of the root system architecture and its plasticity in drought conditions has been documented in major legumes. QTLs for root surface area [30] and root length [31] in soybean, root diameter in cowpea [34], root length in pea [35], basal root angle in common bean [36], and rooting depth, root surface area and root length density in chickpea [37] and root surface area, lateral root number and specific root length [118] in lentil have been reported. For other essential crops such as Rice [119, 120], durum wheat [121], barley [122, 123], maize [124], sorghum [125], pearl millet [126], finger millet [111], and cotton [127] very significant progress has been made in the understanding and use of root traits in breeding programs. The ideal root architectural ideotype for abiotic stress tolerance for optimized nutrient and water acquisition and even coining the phrase ' . . .steep, cheap and deep . . .' [128] i.e. steep (root angle) [129], cheap (metabolic costs) [130] and deep (root architectural arrangements) [46, 131]. There is ample evidence that the entries chosen on the basis of semi-hydroponics or hydroponics are also successful in soil and field experiments [132–135].

Mungbean genotypes identified in this experiment with such a wide variety of root properties could be used for subsequent studies in greenhouses and on-field assessment. Finally, the development of mapping population, use of molecular markers technology, root simulations, and gene mapping to develop germplasm with improved root traits for better tolerance to water deficit and harsh conditions.

## Supporting information

**S1 Table. Passport data of the AVRDC mungbean mini core collection.**
(DOCX)

**S2 Table. Mean scores of the AVRDC mungbean mini core collection for fourteen root traits.**
(DOCX)

## Author Contributions

**Conceptualization:** Muraleedhar S. Aski,  Gayacharan, Renu Pandey, Ramakrishnan M. Nair.

**Data curation:** Muraleedhar S. Aski, Gyan Prakash Mishra, Dharmendra Singh.

**Formal analysis:** Muraleedhar S. Aski, Neha Rai, Venkata Ravi Prakash Reddy.

**Funding acquisition:** Ramakrishnan M. Nair, Roland Schafleitner.

**Investigation:** Muraleedhar S. Aski, Harsh Kumar Dikshit, Arun Kumar, Madan Pal Singh.

**Methodology:** Muraleedhar S. Aski, Renu Pandey, Roland Schafleitner.

**Project administration:** Muraleedhar S. Aski.

**Resources:**  Gayacharan, Gyan Prakash Mishra, Arun Kumar, Renu Pandey, Madan Pal Singh, Aditya Pratap, Ramakrishnan M. Nair, Roland Schafleitner.

**Software:** Venkata Ravi Prakash Reddy, Harsh Kumar Dikshit.

**Supervision:**  Gayacharan, Arun Kumar, Ramakrishnan M. Nair.

**Validation:** Neha Rai,  Gayacharan, Roland Schafleitner.

**Visualization:** Dharmendra Singh, Arun Kumar, Aditya Pratap, Ramakrishnan M. Nair.

**Writing – original draft:** Muraleedhar S. Aski.

**Writing – review & editing:** Ramakrishnan M. Nair, Roland Schafleitner.

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
