## [Decision Letter · Decision Letter 0]

28 Oct 2020

PONE-D-20-21922

Characterizing root traits in mungbean mini-core collection (MMC) from World Vegetable Center (AVRDC) Taiwan

PLOS ONE

Dear Dr. Aski,

Thank you for submitting your manuscript to PLOS ONE. After careful consideration, we feel that it has merit but does not fully meet PLOS ONE’s publication criteria as it currently stands. Therefore, we invite you to submit a revised version of the manuscript that addresses the points raised during the review process.

Please address reviewer comments and submit revised manuscript. 

We look forward to receiving your revised manuscript.

Kind regards,

Dorin Gupta, Ph.D.

Academic Editor

PLOS ONE

Journal Requirements:

2. Our internal editors have looked over your manuscript and determined that it is within the scope of our Call for Papers on Plant Phenomics & Precision Agriculture. This collection of papers is headed by a team of Guest Editors for PLOS ONE.

Additional information can be found on our announcement page: https://plos.io/phenomics

If you would like your manuscript to be considered for this collection, please let us know in your cover letter and we will ensure that your paper is treated as if you were responding to this call.

If you would prefer to remove your manuscript from collection consideration, please specify this in the cover letter.

3. Please amend the manuscript submission data (via Edit Submission) to include author Gayacharan.

4. Please upload a new copy of Figure 6 as the detail is not clear. Please follow the link for more information: https://blogs.plos.org/plos/2019/06/looking-good-tips-for-creating-your-plos-figures-graphics/

5. We note you have included a table to which you do not refer in the text of your manuscript. Please ensure that you refer to Table 4 in your text; if accepted, production will need this reference to link the reader to the Table.

6. Please include captions for your Supporting Information files at the end of your manuscript, and update any in-text citations to match accordingly. Please see our Supporting Information guidelines for more information: http://journals.plos.org/plosone/s/supporting-information

Reviewers' comments:

Reviewer's Responses to Questions

**Comments to the Author**

1. Is the manuscript technically sound, and do the data support the conclusions?

Reviewer #1: Yes

Reviewer #2: Yes

2. Has the statistical analysis been performed appropriately and rigorously? 

Reviewer #1: Yes

Reviewer #2: Yes

3. Have the authors made all data underlying the findings in their manuscript fully available?

Reviewer #1: Yes

Reviewer #2: Yes

4. Is the manuscript presented in an intelligible fashion and written in standard English?

Reviewer #1: Yes

Reviewer #2: Yes

5. Review Comments to the Author

Reviewer #1: PONE-D-20-21922: Characterizing root traits in mungbean mini-core collection (MMC) from World Vegetable Center (AVRDC) Taiwan.

Main focus of this MS is to identify phenotypic differences in the root system architectural traits and correlations of key root traits, towards mung bean breeding for more efficient nutrient and water uptake for better management of abiotic stresses. The experiments are focused and well done. Screening for natural genetic variations in root architectural traits provides key information towards breeding programs for better plant performances in mung bean. This MS is suitable for publication in PLOS One after addressing the following minor comments:

Line 19 – root key root traits change to key root traits

Line 56 – Modify the sentence

Introduction: Add some details showing the updates/stubs of root system architecture analysis and application of root traits in legume improvement (there are several studies in soybean, chickpea and other legumes).

Discussion: This section need to be improved. At present it is descriptive with repeating contents from the results section. Need more supporting evidences and take home messages based on the findings.

Also, it will be good if the authors can check/edit the entire MS documents for English language and grammar usage.

Reviewer #2: Line 40: Per hectare?

Line 42: Remove "that result in reduced grain yields"

Line 43-51: This is far fetched and more generalized or vague. More evidence needed to show that effects of changes in global temperatures actually impacted Mungbean production.

Line 48-51: Provide references for each of the factor mentioned here that affects Mung bean production.

Line 53: Remove "Crop breeding programmes mainly have focused on above-ground organs".

Line 56-57: Would expect a brief description on some screening techniques that are being developed and proven useful at least latest. Also a sentence or two is needed to know why earlier methods weren't useful.

Line 58-61: How does it connect with the preceding statements on Screening techniques, does the author used new techniques to report the findings what he did?

Line 62: A brief description of ideal root architecture attributes is needed.

Line 76-78: Introduction is plagued with more generalised statements and doesn't discuss the topic in depth, the topic is about Mungbean but not a sentence shows what issues are with Mungbean root adaptation and why one should explore a global germplasm to find variation for roots. The next important theme of this study is root phenotyping platforms, there are many statements about this but not exclusive or conclusive explanation provided, therefore the introduction may need restructuring inclusive of these topics to ensure flow with the references.

Line 84: In field or glasshouse?

Line 85: Is it Conviron growth chamber, if so or not please provide details of the growth chamber.

Line 86: move the abbreviation to line 84

Line 95: Please add from where you purchased these items, germination stand, trays, germination paper.

Line 98: Cannot resolve how you managed three replications here, 9*3*12=324, Total number of accessions are 296 (line 83) and 3 replications of each would make = 888 accessions. Please clarify.

Line 111: Any version number or manufacturer details.

Line 115: Replace the title with Assessment of root phenotypes.

Line 126: please provide instrument details

Line 145: Material and methods lacks bit of clarity on how the experiments were conducted particularly information on replications is ambiguous. Further lacks the description of the process of how the measurements were undertaken. There are different equipment used in this study but none of them have manufacture details.

Line 201-206: Topic for Discussion section

Line 208-215: This paragraph is not necessary, you may start with your aim and focus on the main outcomes of the study.

Line 226: Although few references are listed, the section does not provide or discuss the disadvantages of previous methods and a comparison with proper examples for both methods needed to make it more convincing.

Line 228: Any reference?

Line 232: What does this mean, helpful or not useful for benefit of mungbeans?

Line 242: examples from any pulse crop may further strengthen your statements.

Line 246: Again this is another generalized statement, please briefly state what it means as many readers would be interested.

Line 277: Discussion needs bit more on mungbean literature about which root traits were previously researched and whether selection for those traits proved beneficial or failure. Report some examples of existing varieties that are susceptible to abiotic stress and where root parameters were considered as a reason. This information may provide to convince readers why you may need to explore root variation in a global germplasm. I'm convinced that this study is not just to test a new screening method in mungbeans. Although there are some good statistical classifications evaluated to divide the variation into clusters, it would be good to provide a list of lines with ideal root phenotypes and also provide a comment on undesirable phenotypes.

Line 324: Make it italics and follow same all through

Line 334: Be consistent, either use abb. or full forms or follow journal guidelines

Line 562: why only 3 plants tested when 10 plants were actually used.

6. PLOS authors have the option to publish the peer review history of their article (what does this mean?). If published, this will include your full peer review and any attached files.

Reviewer #1: No

Reviewer #2: No

---

## [Author Response · Author response to Decision Letter 0]

12 Dec 2020

As per editor suggestions corrections are carried to meet Journal Requirements:

1. PLOS ONE's style requirements, including those for file naming. 

2. Remove our manuscript from collection consideration.

3. Included author Gayacharan.

4. new copy of Figure 6is uploaded

5. Table 4 in text was included

 6. Captions in Supporting Information files included 

Response to reviewers is uploaded separate file

---

## [Decision Letter · Decision Letter 1]

20 Jan 2021

PONE-D-20-21922R1

Assessment of root phenotypes in mungbean mini-core collection (MMC) from the World Vegetable Center (AVRDC) Taiwan

PLOS ONE

Dear Dr. Aski,

Thank you for addressing reviewer comments in your first revised and submitted manuscript to PLOS ONE. After careful consideration, we feel that it has merit but does not fully meet PLOS ONE’s publication criteria as it currently stands. Therefore, we invite you to submit a revised version of the manuscript that addresses the minor points raised during the review process.

We look forward to receiving your revised manuscript.

Kind regards,

Dorin Gupta, Ph.D.

Academic Editor

PLOS ONE

Additional Editor Comments (if provided):

Please address second reviewers comments and submit revised manuscript.

Reviewers' comments:

Reviewer's Responses to Questions

**Comments to the Author**

1. If the authors have adequately addressed your comments raised in a previous round of review and you feel that this manuscript is now acceptable for publication, you may indicate that here to bypass the “Comments to the Author” section, enter your conflict of interest statement in the “Confidential to Editor” section, and submit your "Accept" recommendation.

Reviewer #1: (No Response)

Reviewer #2: All comments have been addressed

2. Is the manuscript technically sound, and do the data support the conclusions?

Reviewer #1: Yes

Reviewer #2: Yes

3. Has the statistical analysis been performed appropriately and rigorously? 

Reviewer #1: Yes

Reviewer #2: Yes

4. Have the authors made all data underlying the findings in their manuscript fully available?

Reviewer #1: Yes

Reviewer #2: Yes

5. Is the manuscript presented in an intelligible fashion and written in standard English?

Reviewer #1: Yes

Reviewer #2: Yes

6. Review Comments to the Author

Reviewer #1: (No Response)

Reviewer #2: Thanks for addressing all the comments made. Well done!

Here are few more minor comments.

Line 150 : Strongly recommend inclusion of technical replicates (at least the best performing accessions) for such studies to enhance statistical significance

Line 221 : TDW instead of TWD??

Line 222 : Could this line be modified? At the moment, this doesn't align well with the next statement.

Line 379 : Consider adding a sentence on a successful on - field establishment of derivatives from some of the previous studies using semi-hydroponic method if available.

Table 2 : Add collection

Also add AVRDC before Mungbean mini core collection throughout the manuscript.

Table 3 : Repeat above

Table 5 : Semi-hydroponic conditions

Figure 1 : Germination stand?

Also it would be ideal to add a picture of plastic tray mentioned in M & M, the order can be growth chamber (a), plastic tray (b), germination stand (c), and the rest follow in the order they are.

Figure 2 : add "of" in between distribution and root traits.

It should be number of accessions on the y- axis rather than frequency

Figure 6 : Will recommend to add cluster names as colour legends for this dendrogram, can be made in MS powerpoint

Title for S1 Table : Passport data of AVRDC Mungbean mini core collection

Title for S2 Table : Mean scores of AVRDC Mungbean mini core collection for fourteen root traits

7. PLOS authors have the option to publish the peer review history of their article (what does this mean?). If published, this will include your full peer review and any attached files.

Reviewer #1: No

Reviewer #2: No

---

## [Author Response · Author response to Decision Letter 1]

2 Feb 2021

Few more minor comments Reply 

1 Line 150 : Strongly recommend inclusion of technical replicates (at least the best performing accessions) for such studies to enhance statistical significance: 

Reply: Technical replicates was included instead of biological replicates

2 Line 221 : TDW instead of TWD??: Reply :Yes, it is TDW. 

3 Line 222 : Could this line be modified? At the moment, this doesn't align well with the next statement. Reply : Yes, line was modified and rewritten as While RDW showed positive association with other biomass traits like SDW and TDW.

4 Line 379 : Consider adding a sentence on a successful on - field establishment of derivatives from some of the previous studies using semi-hydroponic method if available. : 

Reply : Added as There is ample evidence that the entries chosen on the basis of semi-hydroponics or hydroponics are also successful in soil and field experiments.[132-135]

Chen YL, Dunbabin VM, Postma JA, Diggle AJ, Palta JA, Lynch JP, Siddique KH, Rengel Z. Phenotypic variability and modelling of root structure of wild Lupinus angustifolius genotypes. Plant and Soil. 2011 Nov;348(1):345-64.

Baier AC, Somers DJ, Gusiafson JP. Aluminium tolerance in wheat: correlating hydroponic evaluations with field and soil performances. Plant Breeding. 1995 Aug;114(4):291-6.

Gahoonia TS, Nielsen NE. Barley genotypes with long root hairs sustain high grain yields in low-P field. Plant and Soil. 2004 May;262(1):55-62.

Tavakkoli E, Fatehi F, Rengasamy P, McDonald GK. A comparison of hydroponic and soil-based screening methods to identify salt tolerance in the field in barley. Journal of experimental botany. 2012 Jun 13;63(10):3853-67.

5 Table 2 : Add collection Also add AVRDC before Mungbean mini core collection throughout the manuscript.: Reply: Corrected throughout the manuscript.

6 Table 3 : Repeat above Reply: Yes repeated as suggested

7 Table 5 : Semi-hydroponic conditions Reply: Yes corrected as suggested

8 Figure 1: Germination stand? Also it would be ideal to add a picture of plastic tray

mentioned in M & M, the order can be growth chamber (a), plastic tray (b), germination stand (c), and the rest follow in the order they are. Reply: Corrected as per sequence suggested 

order can be (a) growth chamber, (b) plastic tray, (c) germination stand 

9 Figure 2 : add "of" in between distribution and root traits. It should be number of accessions on the y- axis rather than frequency 

Reply: Edited and corrected as per suggestions

10 Figure 6 : Will recommend to add cluster names as colour legends for this dendrogram, can be made in MS powerpoint 

Reply: Edited and corrected as per suggestions

11 Title for S1 Table : Passport data of AVRDC Mungbean mini core collection: 

Reply: Edited and corrected as per suggestions

12 Title for S2 Table : Mean scores of AVRDC Mungbean mini core collection for fourteen root traits : Reply: Edited and corrected as per suggestions

---

## [Editor Report · Decision Letter 2]

16 Feb 2021

Assessment of root phenotypes in mungbean mini-core collection (MMC) from the World Vegetable Center (AVRDC) Taiwan

PONE-D-20-21922R2

Dear Dr. Aski,

We’re pleased to inform you that your manuscript has been judged scientifically suitable for publication and will be formally accepted for publication once it meets all outstanding technical requirements.

Kind regards,

Dorin Gupta, Ph.D.

Academic Editor

PLOS ONE
---

## [Editor Report · Acceptance letter]

22 Feb 2021

PONE-D-20-21922R2 

Assessment of root phenotypes in mungbean mini-core collection (MMC) from the World Vegetable Center (AVRDC) Taiwan 

Dear Dr. Aski:

I'm pleased to inform you that your manuscript has been deemed suitable for publication in PLOS ONE. Congratulations! Your manuscript is now with our production department. 

Kind regards, 

on behalf of

Dr. Dorin Gupta 

Academic Editor

PLOS ONE